# Using Normalized Carcinoembryonic Antigen and Carbohydrate Antigen 19 to Predict and Monitor the Efficacy of Neoadjuvant Chemotherapy in Locally Advanced Gastric Cancer

**DOI:** 10.3390/ijms241512192

**Published:** 2023-07-29

**Authors:** Xiao-Huan Tang, Xiao-Long Wu, Xue-Jun Gan, Yi-Ding Wang, Fang-Zhou Jia, Yi-Xue Wang, Yan Zhang, Xiang-Yu Gao, Zi-Yu Li

**Affiliations:** 1Key Laboratory of Carcinogenesis and Translational Research (Ministry of Education), Department of Gas-Trointestinal Cancer Center, Ward I, Peking University Cancer Hospital & Institute, Beijing 100142, China; tangxiaohuan@bjmu.edu.cn (X.-H.T.); kimi112358wxl@hotmail.com (X.-L.W.);; 2Biological Sample Bank, Peking University Cancer Hospital & Institute, Beijing 100142, China

**Keywords:** locally advanced gastric cancer, neoadjuvant chemotherapy, normalization of CEA/CA19-9, treatment efficacy, overall survival

## Abstract

Carcinoembryonic antigen (CEA) and carbohydrate antigen 19-9 (CA19-9) are established prognostic biomarkers for patients with gastric cancer. However, their potential as predictive markers for neoadjuvant chemotherapy (NACT) efficacy has not been fully elucidated. Methods: We conducted a retrospective analysis to determine values of CEA and CA19-9 prior to NACT (pre-NACT) and after NACT (post-NACT) in 399 patients with locally advanced gastric cancer (LAGC) who received intended NACT and surgery. Results: Among the 399 patients who underwent NACT plus surgery, 132 patients (33.1%) had elevated pre-NACT CEA/CA19-9 values. Furthermore, either pre-NACT or post-NACT CEA /CA19-9 levels were significantly associated with prognosis (*p* = 0.0023) compared to patients with non-elevated levels. Moreover, among the patients, a significant proportion (73/132, 55.3%) achieved normalized CEA/CA19-9 following NACT, which is a strong marker of a favorable treatment response and survival benefits. In addition, the patients with normalized CEA/CA19-9 also had a prolonged survival compared to those who underwent surgery first (*p* = 0.0140), which may be attributed to the clearance of micro-metastatic foci. Additionally, the magnitude of CEA/CA19-9 changes did not exhibit a statistically significant prognostic value. Conclusions: Normalization of CEA/CA19-9 is a strong biomarker for the effectiveness of treatment, and can thus be exploited to prolong the long-term survival of patients with LAGC.

## 1. Introduction

Gastrectomy plus perioperative chemotherapy, including neoadjuvant and adjuvant chemotherapy, is the dominant treatment for patients with locally advanced gastric cancer (LAGC). Neoadjuvant chemotherapy (NACT) has several potential advantages, including tumor down-staging, increasing the rate for curative resection, clearing micro-metastases, reducing locoregional recurrence and providing drug sensitivity information [1,2]. Clinical results have demonstrated that NACT combined with surgery can significantly prolong the overall survival (OS) and disease-free survival (DFS) of patients with LAGC [3,4]. In addition, recent findings from multiple clinical trials have also confirmed the survival benefits of NACT [5,6,7], establishing it as the standard therapeutic strategy for patients with LAGC in certain regions of the Asia-Pacific and the Western hemisphere [2,3,5,8,9,10].

To date, however, a significant proportion of LAGC patients remain insensitive to NACT [11]. For example, results from a meta-analysis indicated that the average rate of a pathological complete response (pCR), a promising surrogate of survival [12,13], was only 6.74% (ranging from 3 to 15%) following NACT in patients with LAGC [14]. Approximately half of the patients did not show an effective response to NACT. Therefore, there is a need to enhance the individualized precision of NACT by exploring novel strategies [2]. Therefore, it is crucial to screen patients who are sensitive to NACT and explore individualized precision strategies for NACT in order to improve survival outcomes for these patients.

Although various approaches for predicting NACT efficacy in LAGC patients have been evaluated in the past decades [15,16], optimal models with a good predicting value are still lacking. Carcinoembryonic antigen (CEA) and carbohydrate antigen 19-9 (CA19-9), the most widely used quantitative prognostic biomarkers in LAGC [17], have been shown to be independent prognostic factors for LAGC patients. Additionally, previous studies [17] also found that a positive pretreatment level of CEA/CA19-9 was an indicator of a poor prognosis in LAGC patients, and a significant decrease in tumor markers was observed after NACT. In addition, the changes in tumor markers may be a common characteristic in many types of tumors, as a decrease in tumor marker levels after chemotherapy has also been reported in pancreatic adenocarcinoma [18], advanced cholangiocarcinoma [19], locally advanced rectal cancer [20], ovarian cancer [21] and metastatic colorectal cancer [22,23]. Although most studies suggest that a decrease in CA125 levels indicates a favorable response to chemotherapy in ovarian cancer [24], Coleman and colleagues discovered that serum CA125 concentration typically rises after the first and second cycles of chemotherapy and subsequently decreases [21]. In addition, Tsai et al. revealed that normalization of CA19-9 following NACT is a prognostic marker for survival in pancreatic adenocarcinoma [18]. However, the correlation between dynamic changes in CEA/CA19-9 and the efficacy of NACT in LAGC has not been well analyzed.

In the present study, we analyzed changes in CEA and CA19-9 levels in patients with LAGC who received NACT and D2 gastrectomy, and found that normalization of CEA or/and CA19-9 following NACT suggested a good efficacy and could be used to guide neoadjuvant therapy.

## 2. Results

### 2.1. Patient Characteristics

A total of 399 patients with LAGC who underwent NACT followed by surgery were included in this study. Preoperative and postoperative data for CEA and CA19-9 were available for analysis. All patients received intended NACT and D2 gastrectomy plus lymphadenectomy from January 2007 to June 2016 in Peking University Cancer Hospital (PKUCH). Among them, 132 (33.1%) had elevated CA19-9 values (>37.0 U/mL) or/and elevated CEA values (>5.0 U/mL). The median values prior to NACT (pre-NACT) CEA and CA19-9 were 2.26 U/mL and 10.37 U/mL for all included patients (Table 1). Details of clinical–pathological characteristics of the 399 enrolled patients are shown in Table 1. In sum, 98 (24.6%) and 301 (75.4%) of the patients were female and male, respectively. All patients had a median age of 61.0 (21.0~86.0) years, while those with normal and elevated marker values were 60.0 (21.0~79.0) and 64.0 (36.0~86.0), respectively. The median body mass index (BMI) value was 23.69 kg/m2. Moreover, 316 (79.2%) patients were at clinic stage III, while 72 (18.0%) and 11 (2.8%) were at stages II and IV, respectively. Patients with elevated CEA/CA19-9 levels had a higher age (*p* = 0.026) and more advanced clinical N (*p* = 0.002) and tumor node metastasis (TNM) stages (*p* = 0.002), and no significant differences were observed in other characteristics between the groups.

### 2.2. Neoadjuvant Therapy

Details of the NACT regimen and cycles are also provided in Table 1. The enrolled patients were treated with different fluorouracil plus platinum regimens, including SOX in 191 (47.9%) patients, mFOLFOX7 in 84 (21.1%) patients, XELOX in 72 (18.0%) patients and other regimens in 52 (13.0%) patients. All patients had a median cycle of 2.85 (range: 2.00~6.00) for NACT, while those with normal and elevated pre-NACT CEA/CA19-9 levels recorded means of 2.75 and 2.95, respectively.

Additionally, we found no significant differences in surgical settings between patients with normal and elevated pre-NACT CEA/CA19-9 levels (Table 1). Total gastrectomy was performed in 183 (45.9%) patients, distal gastrectomy in 176 (44.1%) patients and proximal gastrectomy in 40 (10.0%) patients. The median number of resected lymph nodes for each patient was 29.

### 2.3. Treatment Outcomes

After NACT, 29 (7.3%) patients exhibited a pathological complete response (pCR, TRG 0), while 57 (14.3%), 106 (26.6%) and 207 (51.9%) patients had TRGs 1, 2 and 3, respectively. The rate of a major pathological response (MPR), consisting of TRGs 0 and 1, was 21.6% in all patients. However, this rate was not significantly associated with pre-NACT CEA/CA19-9 status (*p* = 0.175).

All 399 patients had a median OS of 113.3 months, while those in the pre-NACT normal and elevated CEA or/and CA19-9 groups had a median OS of 126.1 and 40.9 months, respectively. The 5-year OS rates were 62.5% and 49.2% in the pre-NACT normal and elevated CEA or/and CA19-9 groups. Thus, patients with normal pre-NACT CEA or/and CA19-9 had a longer OS than those with elevated CEA/CA19-9 levels (Figure 1A, *p* = 0.0023). However, there were no significant differences between patients with single- and double-positive markers (Figure 1B, *p* = 0.77). Furthermore, the OS of patients with elevated CEA or CA19-9 levels also had no statistic difference (Figure 1C, *p* = 0.81). Similarly, we also analyzed the correlation between OS and CEA and CA19-9 values after NACT (post-NACT), and the results are consistent with the results for before NACT (Figure 1D,F).

A total of 73 (55.3%) out of the 132 patients with elevated pre-NACT CEA/CA19-9 achieved normalization of post-NACT CEA/CA19-9 following NACT. We found no statistically significant differences in clinicopathological information between patients with normalized and non-normalized post-NACT CEA/CA19-9 (Appendix A). In fact, only 4 (3.0%) patients exhibited more elevated CEA or/and CA19-9 levels following NACT in these patients, while only 11 (4.1%) out of the 267 patients with normal pre-NACT CEA and CA19-9 exhibited elevated CEA or/and CA19-9 following NACT. Thus, from the perspective of tumor markers, NACT would not be expected to adversely affect survival.

### 2.4. Prognostic Value of Normalization of Post-NACT CEA/CA19-9

We then analyzed the prognostic value of CEA and CA19-9 normalization. Both CEA and CA19-9 post-NACT normalization indicated a better prognosis in LAGC patients, while CA19-9 exhibited a better predictive value (Appendix A, CEA *p* = 0.1, CA19-9 *p* = 0.025) regardless of the status of another marker, and CEA exhibited a better predictive value (Appendix A, CEA *p* = 0.048, CA19-9 *p* = 0.25) only when the negative status of another marker was included (CEA^+^CA199^−^ and CEA^−^CA19-9^+^) at baseline.

Then, we performed combined analyses and stratified all patients into four groups according to CEA and CA19-9 levels. These groups comprised 260 N-N (from normal pre-NACT CEA/CA19-9 to normal post-NACT CEA/CA19-9), 73 E-N (from elevated pre-NACT CEA/CA19-9 to normal post-NACT CEA/CA19-9), 11 N-E (from normal pre-NACT CEA/CA19-9 to elevated post-NACT CEA/CA19-9) and 55 E-E (from elevated pre-NACT CEA/CA19-9 to elevated post-NACT CEA/CA19-9). The median OS for patients in the N-N, N-E and E-E groups was 113.3, 26.8 and 20.5 months, respectively, while no median OS was recorded in the E-N group (Figure 2A, *p* < 0.0001). Patients in the E-N group had a prolonged OS compared to the E-E group (Figure 2B, *p* = 0.0062). Furthermore, the survival difference between pre-NACT positive- and negative-CEA/CA19-9 patients was eliminated when markers’ normalization was achieved (Figure 2C, N-N vs. E-N *p* = 0.491). However, there were also no significant differences in the OS between patients with single-marker normalization and double-marker normalization (Figure 2D, *p* = 0.950), and both of the above had survival benefits compared to non-normalization (Figure 2E,F, *p* = 0.130 and *p* = 0.010). Therefore, normalization of post-NACT CEA or/and CA19-9 is a good predictor of therapeutic efficacy and patients’ OS, either for one or two markers’ normalization.

Additionally, results from univariable Cox proportional hazard regression analyses show that patients with post-NACT CEA/CA19-9 normalization had a 0.52-fold (Appendix A, 95%CI: 0.32–0.84, *p* = 0.007) decreased risk of death, compared to those without normalization. Multivariable Cox proportional hazard regression analyses revealed that post-NACT CEA/CA19-9 normalization was an independent prognostic biomarker (Appendix A, *p* = 0.039).

On the other hand, the rates of MPR in the N-N, E-N, N-E and E-E groups were 21.5, 31.5, 0 and 11.9%, respectively (Figure 3A). Notably, patients who achieved normalization of CEA/CA19-9 after NACT exhibited a higher MPR rate compared to those without normalization (Figure 3B, 31.5% vs. 11.9%, *p* = 0.014). However, results from correlation analysis between NACT cycles and post-NACT CEA/CA19-9 normalization revealed that more extensive treatment was not significantly associated with either normalization of post-NACT CEA/CA19-9 (Figure 3C, *p* = 0.221) or a longer OS of LAGC patients (Figure 3D, *p* = 0.528). Thus, increasing the rate of NACT for patients who did not achieve CEA/CA19-9 normalization after 2-3 cycles of NACT needs to be considered carefully according to the patients’ situation.

### 2.5. Patients with Elevated CEA/CA19-9 Levels and Normalization following NACT Had a Survival Benefit

To further verify the survival benefit of NACT-induced CEA/CA19-9 normalization, we analyzed a total of 246 LAGC patients with elevated baseline CEA/CA19-9 who did not receive NACT prior to surgery. The aim was to determine whether NACT-induced downregulation of CEA/CA19-9 is correlated to survival benefits. Results show that patients with normalized CEA/CA19-9 had a better prognosis (Figure 4A, *p* = 0.014). The median OS for patients in the non-NACT group was 32.2 months, while this was not recorded in the E-N group (Figure 4A). Furthermore, there were no significant differences between the OS of patients in the non-NACT and E-E groups (Figure 4B, *p* = 0.45). These results reveal that LAGC patients with elevated CEA/CA19-9 levels and normalization after NACT had a survival benefit.

### 2.6. The Magnitude of Change in CEA/CA19-9 Levels and OS

Next, we calculated changes in CEA/CA19-9 during NACT to investigate whether survival benefits were associated with the magnitude of CEA/CA19-9 changes in patients with an elevated pre-NACT CEA/CA19-9 value (CEA > 5.0 U/mL; CA19-9 > 37.0 U/mL). After excluding patients with post-NACT CEA/CA19-9 normalization, the remaining patients were separated into two groups based on changes in CEA/CA19-9. We found that the magnitude of CEA/CA19-9 changes could not effectively predict the OS of patients. Patients with decreased CEA/CA19-9 values did not have a better prognosis than patients with increased CEA/CA19-9 values following NACT (Figure 4C,D, CEA *p* = 0.42, CA19-9 *p* = 0.38). Furthermore, we also found no significant differences in OS among patients with different magnitudes of change (decrease > 50% vs. <50%) in CEA and CA19-9 (Figure 4E,F, CEA *p* = 0.82, CA19-9 *p* = 0.26). Therefore, the magnitude of change in CEA/CA19-9 levels was a poor predictor of OS.

### 2.7. Elevated CEA/CA19-9 Predicted a Higher Probability of Metastasis

To better understand the potential mechanisms underlying the improved prognosis associated with the normalization of CEA/CA19-9, further investigations are warranted. Next, we characterized patients with elevated CEA/CA19-9 by analyzing the mRNA expression profiles of tumor tissues before NACT in 63 patients, including 50 and 13 patients with initial normal and elevated CEA/CA19-9, respectively. Then, 170 differentially expressed genes (DEGs) were identified between normal and elevated CEA/CA19-9 tissues, based on a *p* < 0.05 (Figure 5A). Subsequently, the function of these DEGs was analyzed via KEGG enrichment analysis, and the results show that the up-regulated genes in tissues with elevated CEA/CA19-9 were associated with gastric carcinogenesis and several signaling pathways regulating metastasis, such as that regulating pluripotency of stem cells [25], the Wnt pathway [26], the mTOR signaling pathway [27] and the Hippo signaling pathway (Figure 5B) [28]. Conversely, the corresponding downregulated genes were involved in the weakening of intercellular cohesion and lowering the level of immunoreactivity (Figure 5C), which may promote immune escape and tumor metastasis. Taken together, these findings suggest that patients with elevated CEA/CA19-9 levels are more likely to have tumor cells that have metastasized, and the survival benefits associated with the normalization of CEA/CA19-9 may be attributed to the clearance of micro-metastases.

## 3. Discussion

We evaluated the prognostic value of changes in CEA and CA19-9, following NACT and D2 gastrectomy, in LAGC patients and found several clinically important aspects. Firstly, patients with an elevated CEA (>5.0 U/mL) or/and CA19-9 (>37.0 U/mL) status, whether at diagnosis or following NACT, had a poor prognosis, which was coupled with a shorter median OS. Secondly, normalization of post-NACT CEA/ CA19-9 indicated a better efficacy, which was accompanied by a favorable OS for LAGC patients. Thirdly, the magnitude of change in CEA/CA19-9 did not have a significant prognostic value in these patients. Fourthly, the RNA-seq results suggest that patients with elevated CEA/CA19-9 are more likely to develop micro-metastases. Fifthly, conducting NACT prior to surgery was recommended in patients with initially elevated CEA/CA19-9, owing to the therapeutic survival benefits achieved with this approach compared to surgery first. Finally, more than half of the patients who initially had elevated CEA/CA19-9 achieved normalization after NACT. Taken together, these findings indicate that the normalization of CEA/CA19-9 is a promising prognostic marker that can be used to guide NACT in patients with LAGC. Based on these findings, we created a flow chart summarizing the process of NACT for patients with LAGC (Figure 6).

Although numerous studies have verified the significant survival benefits of NACT in patients with LAGC [3,5,29], a significant proportion of this group of patients is not sensitive to the therapy [11]. Additionally, currently available biomarkers in clinical practice lack the ability to effectively predict the efficacy of NACT in LAGC. In fact, the current treatment strategies primarily revolve around the clinical stage at first diagnosis. Therefore, the identification of effective, novel and easily accessible predictive and prognostic biomarkers is crucial for achieving individualized precision NACT in LAGC patients and improving their prognosis. Numerous studies have demonstrated that serum tumor markers, especially CEA and CA19-9, can effectively reflect the prognosis of LAGC patients [30,31]. Among these biomarkers, pretreatment CEA levels showed a predictable power for pCR in one study, but these results have not been consistently reproduced in other studies (Appendix A) [17,32,33,34]. In addition, Sun et al. also revealed that the decrease in tumor markers CEA, CA72-4 and CA125 was significant after NACT [17]. However, it is still unknown which specific tumor biomarker or pattern of biomarker changes is most predictive of NACT efficacy. Our results reveal that normalization of CEA/CA19-9 is the strongest predictive marker for treatment response.

Previous studies have correlated initial CEA and CA19-9 levels with important clinicopathological characteristics, such as lymph node metastasis [31], recurrences [35], peritoneal metastasis [36] and survival [37], in GC patients. Notably, NACT can effectively control micro-metastases and locoregional recurrence before surgery. This aspect was corroborated by our RNAseq data from the tumor tissues of patients with elevated and normal CEA/CA19-9, affirming the use of CEA and CA19-9 as clinical markers for detection of micro-metastases. Overall, these results suggest that CEA and CA19-9 may be indicators of NACT’s therapeutic efficacy in GC. However, only a handful of studies have described the role of CEA and CA19-9 levels in the therapeutic sequence or the decision to perform surgical treatment [17]. The results of the present study indicate that patients who initially had elevated CEA/CA19-9 values before NACT and achieved normalization following NACT benefit the most from this treatment approach.

Changes in CEA/CA19-9 levels have been linked to responsiveness to NACT for primary tumors and the potential development of micro-metastases in LAGC patients. Although previous research has demonstrated that elevated CEA/CA19-9 levels following NACT are associated with higher risks of death [17,38], there is no consensus regarding the exact CEA/CA19-9 changes that confer clinical benefits. The results of the present study indicate that the normalization of CEA/CA19-9 following NACT is the best model for indicating clinical benefits, although it may generate modest therapeutic efficacy in the absence of CEA and CA19-9 normalization. This suggests that a tumor-specific response to NACT may not strongly impact survival if normalized CEA/CA19-9 has not been achieved. In addition, functional enrichment analysis of DEGs identified via tissue RNA sequencing suggests that the survival benefits associated with the normalization of CEA/CA19-9 may be attributed to the clearance of potential micro-metastases via NACT. Consequently, dynamic detection of CEA and CA19-9 can be used as a biomarker to guide the application of NACT in LAGC patients. Importantly, CEA and CA19-9 are widely used in the clinical setting as biomarkers and have various advantages, such as their non-invasive nature, easy accessibility and cost-effectiveness. This allows the study findings to be more readily generalized to other centers.

Notably, elevated pre-NACT CEA/CA19-9 levels indicate a higher risk of death for GC patients, although this risk can be reduced through NACT. Normalization of post-NACT CEA/CA19-9 generates an effective therapeutic response, which can improve the OS of patients, while increased post-NACT CEA/CA19-9 values may predict an ineffective treatment. Thus, the recommendation of conducting NACT prior to surgery in patients with initially elevated CEA/CA19-9 levels is of significant importance due to the observed survival benefits compared to the strategy of performing surgery first. By administering chemotherapy before surgery, it is possible to target the tumor and potentially reduce its size, making it more amenable to surgical resection. This sequential treatment strategy has shown promise in improving patient outcomes. In addition to the specific recommendation for patients with elevated CEA/CA19-9 levels, there are other practical implications that should be considered. Firstly, the timing and duration of NACT should be carefully considered. It is important to strike a balance between achieving an adequate response to chemotherapy and minimizing the delay in surgical intervention. Another practical implication is the potential for individualized treatment approaches based on biomarker profiles. By analyzing a combination of biomarkers, including CEA and CA19-9, along with other relevant factors such as tumor stage and molecular characteristics, it may be possible to develop predictive models that can guide treatment decisions and improve patient outcomes.

We subjected the patients enrolled herein to pathological assessment, according to the NCCN guidelines, which is the most commonly used evaluation method for NACT in clinical practice. Although several studies have demonstrated the potential of TRG as a surrogate indicator of patient survival [12,13], this pathological evaluation model considers histological changes in the primary tumor and lymph nodes, which do not reflect changes in micro-metastases. It is possible that this model ignores the survival benefit granted by clearing micro-metastases. Next, we analyzed the OS of patients with elevated CEA/CA19-9 levels following NACT using multivariable Cox regression analyses, and found that the TRG model could not effectively predict prognosis in this population. Furthermore, TRG performed on the resected tumor and lymph node tissues from surgery could not be used to guide therapeutic sequence or the decision to perform the surgical treatment. These findings suggest that dynamic detection of CEA and CA19-9 in LAGC patients is particularly important for guiding treatment and predicting their prognosis.

Although our results indicate that CEA/CA19-9 normalization results in a significant improvement in patient OS following NACT, the study had several limitations. Firstly, this study employed a single-center design, which may have compromised the results. Therefore, multi-center studies are needed to validate our findings. Secondly, the imbalances between NACT regimens and cycles may have induced different CEA/CA19-9 responses, thereby confounding the analysis.

## 4. Materials and Methods

### 4.1. Study Design and Participant Recruitment

We retrospectively analyzed data from a prospectively maintained database comprising 399 LAGC patients who underwent all intended NACT and D2 gastrectomy procedures plus lymphadenectomy between January 2007 and June 2016 at PKUCH. All patients were diagnosed and clinically staged based on endoscopic biopsy specimen analysis, as performed by two pathologists, and contrast-enhanced computed tomography (CT). cTNM and ypTNM stages were determined according to the guidelines of the 8th edition of the TNM grading system [39]. Primary tumors and lymph nodes were deemed to be surgically resectable according to the surgeon’s evaluation. We also collected the clinicopathological characteristics of all patients, including their gender, age, BMI, tumor location, diameter, differentiation, surgical setting, TRG and TNM classification. RNA sequencing (RNA-seq) was performed on specimens from 63 patients (13 and 50 patients with elevated and normal CEA/CA19-9, respectively) before NACT to characterize patterns of mRNA expression. Furthermore, we included a contemporaneous comparison cohort comprising 246 LAGC patients with elevated CEA or/and CA19-9 values, who received surgical treatment first. Patients were followed up every three months for the first two years, and every six months thereafter. The median follow-up time was 49.4 months. OS was defined as survival time since diagnosis. All patients voluntarily signed a written informed consent prior to enrolment in the study, and the study was approved by the Ethics Committee of PKUCH (2019KT05).

### 4.2. Treatment

All patients underwent at least two cycles of NACT with a fluorouracil-based regimen, including SOX, XELOX and FOLFOX, which are recommended by the expert consensus [40]. Radiological evaluation, based on RECIST criteria v1.1 [41], was performed after every 2–3 cycles. Surgery was recommended in cases where the disease had PD or SD, all planned NACT procedures were completed, or the patients could not tolerate or refused NACT. The interval from initiation of NACT to surgery was 3–5 weeks. D2 gastrectomy was performed according to the Guidelines of the Japanese Gastric Cancer Association [42], and the resection range included at least peri-gastric lymphadenectomy with D2 extension and two-thirds of the stomach. Adjuvant chemotherapy was initiated and routinely administered at 3–5 weeks, after the D2 gastrectomy, in patients with an Eastern Cooperative Oncology Group (ECOG) performance status of 0–1. Chemotherapy regimen and cycle were determined according to the therapeutic efficacy of NACT, as well as the T, N, M and TNM stages, which were classified according to the 8th edition of the TNM grading system [39].

### 4.3. CEA and CA19-9 Classification

Levels of CEA and CA19-9 in serum were assessed at two time points, pre-NACT and post-NACT, 3–5 weeks after the completion of all planned NACT procedures. The elevated marker group included LAGC patients with CA19-9 values above 37.0 U/mL and/or CEA values above 5.0 U/mL, which are currently used clinically [19]. The analysis involved several aspects, namely determination of the CA19-9 status pre-NACT, analysis of changes in CEA and CA19-9 in response to NACT ((post-NACT value—pre-NACT value)/ pre-NACT value) and assessment of achievement of a normal post-NACT CEA or/and CA19-9.

### 4.4. Pathological Examination

Regression of tumor grades was performed according to the NCCN guidelines [9] following NACT in patients with LAGC, as follows: grade 0, complete regression with no residual tumor cells; grade 1, near-complete response with single cells or rare small groups of cancer cells; grade 2, partial tumor regression, with residual cancer cells with evident tumor regression, but more than single cells or rare small groups of cancer cells; and grade 3, extensive residual cancer with no evident tumor regression. TRG 0 was defined as pCR, while the MPR was defined as the sum of complete regression and near-complete response.

### 4.5. KEGG Enrichment Analyses

DEGs (*p* < 0.05, fold change > 1.5) were identified using the limma package implemented in R software version 3.6. The KEGG [43] is a database for determining the path of the gene cluster and correlated functions (http://www.genome.jp/kegg/, accessed on 15 March 2022). The KEGG enrichment analysis was performed using the Kobas database (http://kobas.cbi.pku.edu.cn/genelist/, accessed on 5 March 2022) [44] to identify signaling pathways related to the DEGs in KEGG pathways.

### 4.6. Statistical Analyses

Continuous and categorical variables were analyzed using independent samples *t* and chi-square tests, respectively. OS, calculated using the Kaplan–Meier method, was from the initial diagnosis to the date of death or last follow-up, and deaths from all causes were included in the survival analysis. Multiple Cox regression analysis was used to examine the effects of each factor on OS. Statistical analyses were performed using SPSS v25.0 and packages implemented in R software, while data presentation was performed in GraphPad Prism v8.0. *p*-values of less than 0.05 (*p* < 0.05) were considered significant: * *p* < 0.05, ** *p* < 0.01, *** *p* < 0.001, **** *p* < 0.0001.

## 5. Conclusions

The observed changes in CEA and CA19-9 values provide new insights to guide clinicians during evaluation of the complex tumor-specific response to NACT in LAGC patients, and are expected to inform future selection of the optimal therapeutic strategy. Normalization of post-NACT CEA or/and CA19-9 values not only indicates prolonged OS following NACT, but is an independent prognostic predictor of patients with initial elevated CEA/CA19-9. Notably, our results indicate that patients with post-NACT CEA or/and CA19-9 normalization have better OS rates than those who do not receive NACT. Future studies are expected to elucidate whether extended NACT can achieve CEA/CA19-9 normalization.

## Figures and Tables

**Figure 1 ijms-24-12192-f001:**
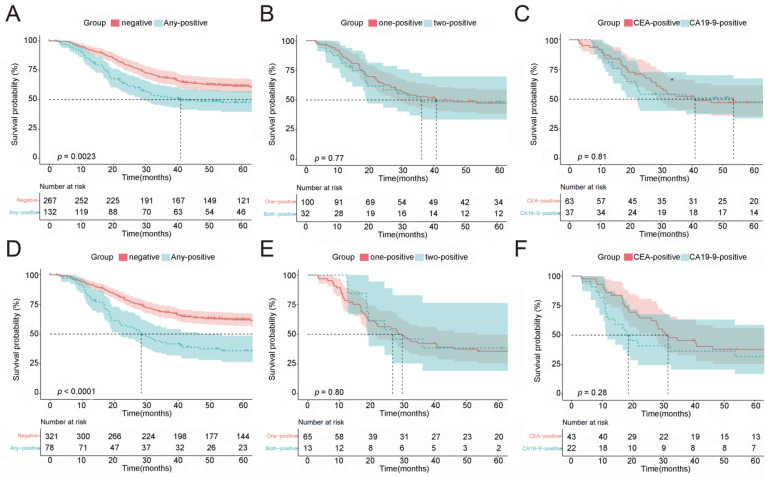
Overall survival of LAGC patients at diagnosis and after NACT based on CEA and CA19-9 values. (**A**) Overall survival by CEA and CA19-9 status prior to NACT: negative: both normal (<5.0 U/mL) CEA and (<37.0 U/mL) CA19-9 levels; any positive: elevated (>5.0 U/mL) CEA or/and (>37.0 U/mL) CA19-9 levels. (**B**) Overall survival of LAGC patients with one elevated marker and both elevated markers. (**C**) Overall survival of LAGC with elevated CEA or CA19-9 levels prior to NACT. (**D**–**F**) Similar analyses of overall survival of LAGC patients after NACT based on CEA and CA19-9 values. LAGC: locally advanced gastric cancer; NACT: neoadjuvant chemotherapy.

**Figure 2 ijms-24-12192-f002:**
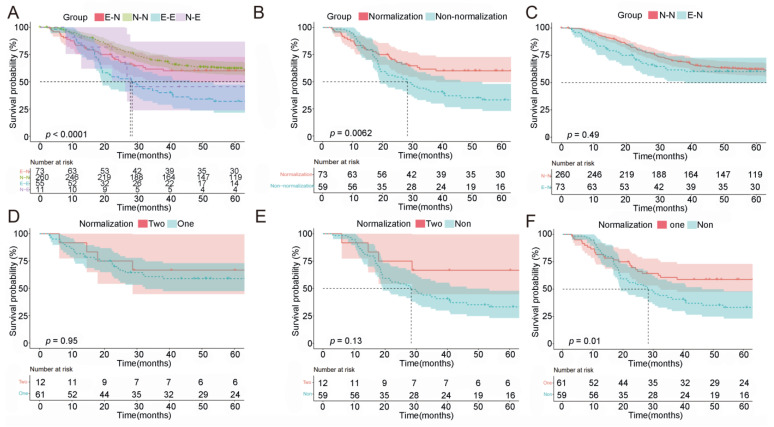
Normalization of CEA/CA19-9 indicated better survival. (**A**) Overall survival by changes in CEA and CA19-9 status. N-N: normal pre-NACT CEA and CA19-9 to normal post-NACT CEA and CA19-9; E-N: elevated pre-NACT CEA/CA19-9 to normal post-NACT CEA/CA19-9; N-E: normal pre-NACT CEA and CA19-9 to elevated post-NACT CEA/CA19-9; E-E: elevated pre-NACT CEA/CA19-9 to elevated post-NACT CEA/CA19-9. (**B**) Overall survival by normalization and non-normalization at CEA/CA19-9 status. (**C**) Overall survival among N-N and E-N patients. (**D**) Overall survival among patients with normalization of two markers (CEA and CA19-9) and one marker (CEA or CA19-9). (**E**) Overall survival among patients with normalization of both CEA and CA19-9 and non-normalization. (**F**) Overall survival among patients with normalization of a single marker of CEA or CA19-9 and non-normalization. NACT: neoadjuvant chemotherapy.

**Figure 3 ijms-24-12192-f003:**
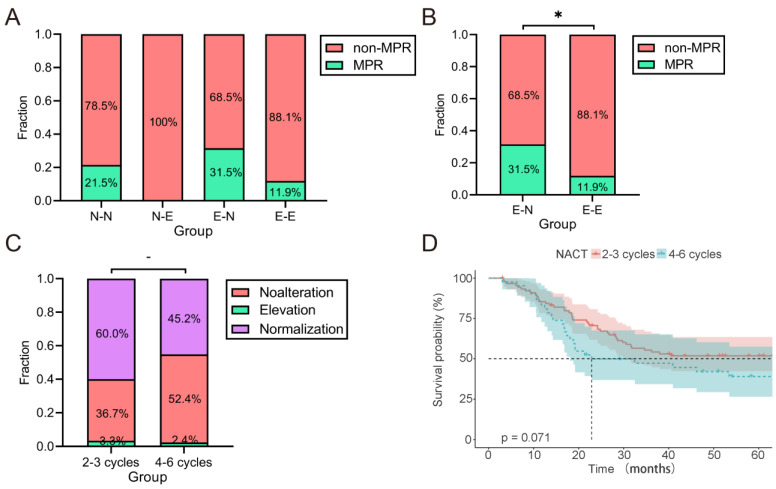
Pathological regression, including MPR (TRG 0 and TRG 1) and non-MPR (TRG 2 and TRG 3), of LAGC patients based on changes in CEA and CA19-9 status. (**A**) The MPR rates among N-N, N-E, E-N and E-E groups. (**B**) The MPR rates among patients with normalization of CEA/CA19-9 (E-N) and non-normalization (E-E) groups. (**C**) The changes in elevated CEA and CA19-9 based on therapeutic intensity: no alteration: no change in number of elevated CEA and CA19-9; elevation: change from normal to elevated CEA/CA19-9; normalization: change from elevated to normal CEA/CA19-9. (**D**) OS among patients treated with NACT for 2-3 cycles and 4-6 cycles. LAGC, locally advanced gastric cancer; MPR, major pathological response; NACT, neoadjuvant chemotherapy; OS, overall survival; TRG: tumor regression grading. * *p* < 0.05.

**Figure 4 ijms-24-12192-f004:**
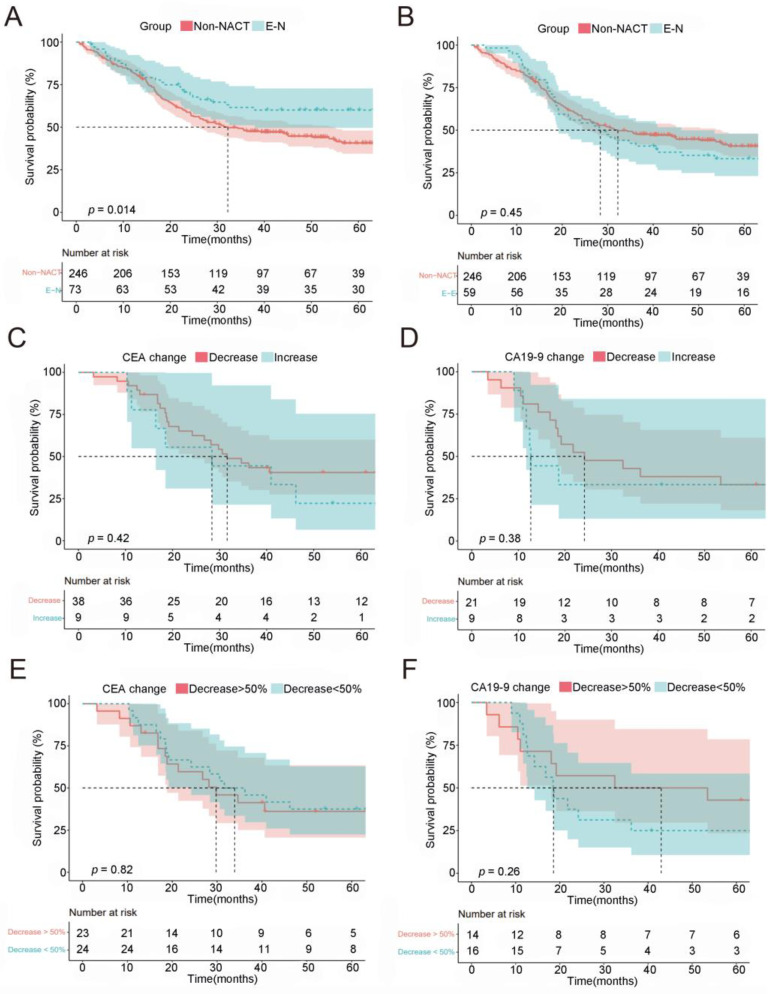
Overall survival by therapeutic strategy among patients with initially elevated CEA/CA19-9 and the magnitude of CEA/CA19-9 changes. (**A**) Overall survival among patients with initially elevated CEA/CA19-9 based on therapeutic strategy: non-NACT: first surgery; E-N: normalization of CEA/CA19-9; (**B**) overall survival by non-NACT and E-E; (**C**,**D**) overall survival by the change in CEA/CA19-9: increased vs. decreased values; (**E**,**F**) overall survival by the magnitude of CEA/CA19-9 changes ((post-NACT value– pre-NACT value)/ pre-NACT value), indicating a >50% decrease and a <50% decrease or increase. NACT, neoadjuvant chemotherapy.

**Figure 5 ijms-24-12192-f005:**
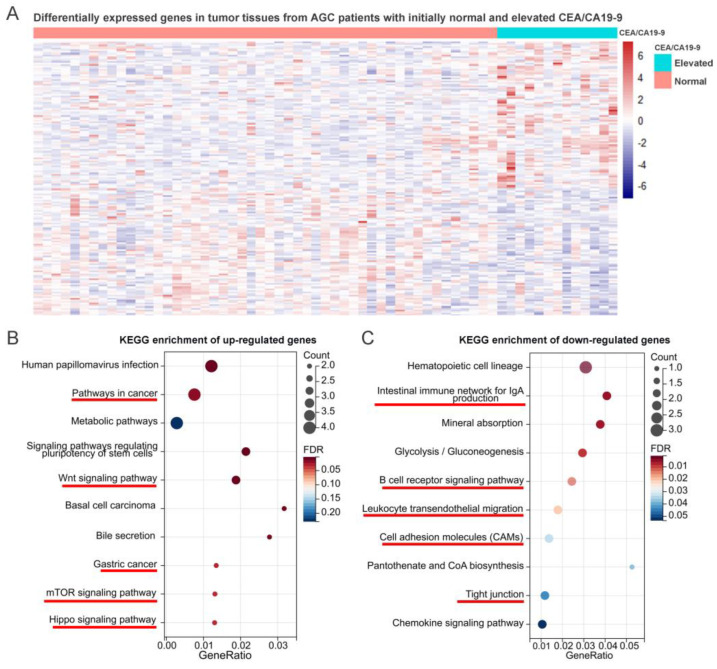
Profiles of mRNA expression in tumor tissues from patients with initially normal and elevated CEA/CA19−9. Differentially expressed genes in tumor tissues from LAGC patients with initially normal and elevated CEA/CA19−9 (**A**). KEGG enrichment results showing enrichment of upregulated (**B**) and downregulated (**C**) genes. LAGC: locally advanced gastric cancer; KEGG: Kyoto Encyclopedia of Genes and Genomes.

**Figure 6 ijms-24-12192-f006:**
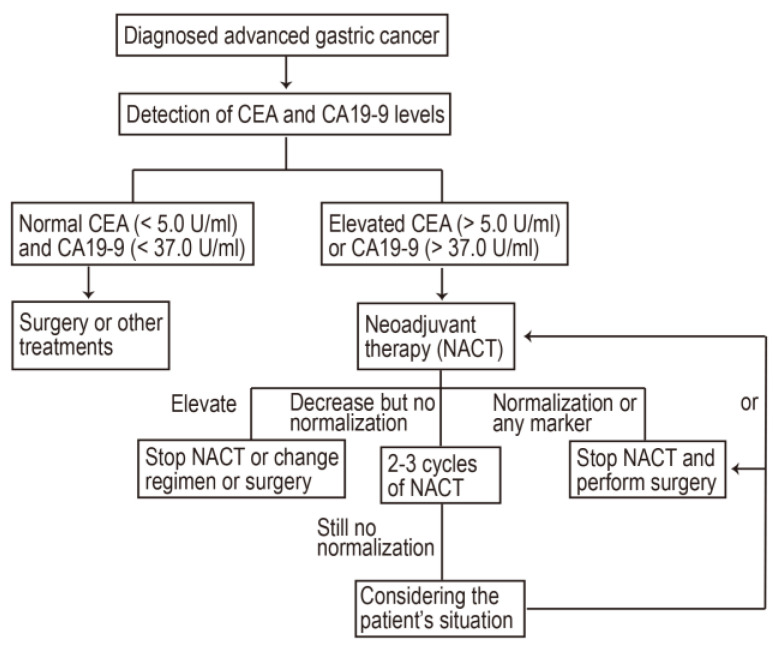
The flow chart of NACT process for patients with LAGC. LAGC, locally advanced gastric cancer; NACT, neoadjuvant chemotherapy.

**Table 1 ijms-24-12192-t001:** Data for clinicopathological information and neoadjuvant chemotherapy.

	All Patients	Without Elevated CEA/CA19-9	With Elevated CEA/CA19-9	*p*-Value
Number of patients	399	267	132	-
Range of CEA value: range	0.2–420.4 U/mL	0.2–4.96 U/mL	0.98–420.4 U/mL	-
Range of CA19-9 value: range	0.6–9867 U/mL	0.6–36.9 U/mL	0.6–9867 U/mL	-
Age in years, (range)	61.0 (21.0~86.0)	60.0 (21.0~79.0)	64.0 (34.0~86.0)	0.026 *
Gender				0.190
Male, n (%)	301 (75.4)	203 (76.0)	98 (74.2)	
Female, n (%)	98 (24.6)	64 (24.0)	34 (21.7)	
BMI (median)	23.69	23.69	23.70	0.775
Location				0.378
Upper, n (%)	141 (35.3)	90 (33.7)	51 (38.6)	
Middle, n (%)	67 (16.8)	47 (17.6)	20 (15.2)	
Lower, n (%)	183 (45.9)	123 (46.1)	60 (45.5)	
Total, n (%)	8 (2.0)	7 (2.6)	1 (0.8)	
Diameter				0.083
<Median, n (%)	203 (50.9)	144 (53.9)	59 (44.7)	
>Median, n (%)	196 (49.1)	123 (46.1)	73 (55.3)	
Differentiation				0.108
Differential, n (%)	212 (53.1)	134 (50.2)	78 (59.1)	
Undifferential, n (%)	187 (46.9)	133 (49.8)	54 (40.9)	
cT stage				0.148
T1-T2, n (%)	6 (1.5)	3 (1.1)	3 (2.3)	
T3, n (%)	158 (39.6)	114 (42.7)	44 (33.3)	
T4, n (%)	235 (58.9)	150 (56.2)	85 (64.4)	
cN stage				0.002 **
N0, n (%)	65 (16.3)	53 (19.9)	12 (9.1)	
N1, n (%)	123 (30.8)	85 (31.8)	38 (28.8)	
N2, n (%)	158 (39.6)	103 (38.6)	55 (41.7)	
N3, n (%)	53 (13.3)	26 (9.7)	27 (20.5)	
cTNM stage				0.002 **
II, n (%)	72 (18.0)	57 (21.3)	15 (11.4)	
III, n (%)	316 (79.2)	207 (77.5)	109 (82.6)	
IV, n (%)	11 (2.8)	3 (1.1)	8 (6.1)	
TRG in primary lesion				0.175
Grade 0	29 (7.3)	19 (7.1)	10 (7.6)	
Grade 1	57 (14.3)	37 (13.9)	20 (15.2)	
Grade 2	106 (26.6)	82 (30.7)	24 (18.2)	
Grade 3	207 (51.9)	129 (48.3)	78 (59.1)	
NACT				0.959
SOX, n (%)	191 (47.9)	127 (47.6)	64 (48.5)	
XELOX, n (%)	72 (18.0)	49 (18.4)	23 (17.4)	
mFOLFOX7, n (%)	84 (21.1)	57 (21.3)	27 (20.1)	
Others, n (%)	52 (13.0)	34 (12.7)	18 (13.6)	
Cycles (median)	2.85	2.75	2.95	0.055
Surgical setting				0.634
Total gastrectomy, n (%)	183 (45.9)	120 (44.9)	63 (47.7)	
Distal gastrectomy, n (%)	176 (44.1)	120 (44.9)	56 (42.4)	
Proximal gastrectomy, n (%)	40 (10.0)	27 (10.1)	13 (8.8)	

BMI, body mass index; NACT, neoadjuvant chemotherapy; TNM, tumor node metastasis; TRG, tumor regression grading. * *p* < 0.05, ** *p* < 0.01.

## Data Availability

The original data presented in this study are included in the article or Appendix A. Requests for other information can be directed to the corresponding authors.

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
