# Peer review of "Using Normalized Carcinoembryonic Antigen and Carbohydrate Antigen 19 to Predict and Monitor the Efficacy of Neoadjuvant Chemotherapy in Locally Advanced Gastric Cancer"

_ijms, 2023, doi:10.3390/ijms241512192_

Round 1

Reviewer 1 Report

This paper states that the decline of the level of serum markers during neoadjuvant therapy for gastric cancer is associated with the improved disease outcome. This is not unexpected: usually, the improvement of many disease characteristics, particularly the ones associated with tumor burden, have positive prognostic value.

The critical deficiency of the paper is the lack of in-depth analysis of the state-of-art. Please carefully consider all studies devoted to the neoadjuvant therapy for gastric cancer, and compare surrogates of the efficacy revealed in these studies. Please analyze, which prognostic markers are reproducible and which are not. I suggest to make a table-format summary for previous studies and current study, and to put this Table into the Supplementary material.

Please review serum marker studies in different tumor types (e.g., CA-125 in ovarian cancer) and conclude whether similar observations (i.e., prognostic value of marker decline) is characteristic for other tumors.

There are potentially important statements, e.g., “Fifthly, NACT prior to surgery was recommended in patients with initially elevated CEA/CA19-9, owing to this therapy’s survival benefits compared to surgery first.” Please discuss this and other practical implications in greater detail.

The section “2.7. Elevated CA19-9 predicted a higher probability of metastasis” is a separate study requiring proper description and discussion.     

The Abstract does not properly reflect the main findings of the paper.

The quality of English is absolutely sufficient for proper understanding of the paper, but some language editing is still required.

The quality of English is absolutely sufficient for proper understanding of the paper, but some language editing is still required.

Author Response

Comments and Suggestions for Authors

This paper states that the decline of the level of serum markers during neoadjuvant therapy for gastric cancer is associated with the improved disease outcome. This is not unexpected: usually, the improvement of many disease characteristics, particularly the ones associated with tumor burden, have positive prognostic value.

Dear reviewer,

First, we would like to express our sincere thanks to you for the constructive and positive comments. We have read and addressed these helpful commons carefully and the details were listed point by point.

The critical deficiency of the paper is the lack of in-depth analysis of the state-of-art. Please carefully consider all studies devoted to the neoadjuvant therapy for gastric cancer, and compare surrogates of the efficacy revealed in these studies. Please analyze, which prognostic markers are reproducible and which are not. I suggest to make a table-format summary for previous studies and current study, and to put this Table into the Supplementary material.

Reply: Thanks very much for your excellent comments. We summarized the research on tumor biomarkers in neoadjuvant chemotherapy for gastric cancer. In addition, we have added a table-format summary of previous studies and current study (Supplementary Table 2) in the discussion section.

“Although numerous studies have demonstrated that serum tumor markers, especially CEA and CA19-9, can effectively reflect the prognosis of LAGC patients, their predictive ability on therapeutic response is limited. Among these biomarkers, pre-treatment CEA levels showed a predictable power for pCR in one study, but the results have not been consistently reproduced in other studies (Supplementary Table 3). In addition, Sun et al. revealed that the decrease of tumor markers CEA, CA72-4, and CA125 was significant after NACT, but further analysis is needed to explore how to utilize this change to guide treatment. Furthermore, it is still unknown which specific tumor biomarker or pattern of biomarker changes is most predictive of NACT efficacy. In this study, our results revealed that normalization of CEA/CA19-9 is the strongest predictive marker for treatment response.”

Please review serum marker studies in different tumor types (e.g., CA-125 in ovarian cancer) and conclude whether similar observations (i.e., prognostic value of marker decline) is characteristic for other tumors.

Reply: Thanks very much for your excellent comments. We have reviewed serum marker studies in different tumor types and found a similar tendency in these markers, such as CEA, CA19-9, CA724 and CA-125, following chemotherapy. The details have been added in the introduction section as follows:

“Previous studies found that pretreatment CEA positive prior to NACT was an indicator for poor prognosis in LAGC patients and a significant decrease of tumor markers was observed after NACT. In addition, the changes in tumor markers may be a common characteristic in many types of tumors, as the decrease in tumor marker levels after chemotherapy has also been reported in pancreatic adenocarcinoma [18], advanced cholangiocarcinoma [19], locally advanced rectal cancer [20], ovarian cancer [21] and metastatic colorectal cancer [22, 23]. Although most studies suggest that a decrease in CA125 levels indicates a favorable response to chemotherapy in ovarian cancer [24], Coleman and colleagues discovered that serum CA125 concentration typically rises after the first and second cycles of chemotherapy and subsequently decreases [21]. In addition, Tsai et al. revealed that normalization of CA19-9 following NACT is a prognostic marker for survival in pancreatic adenocarcinoma [18]. However, the correlation between CEA/CA19-9 dynamic changes and the efficacy of NACT in LAGC has not been well analyzed.”

There are potentially important statements, e.g., “Fifthly, NACT prior to surgery was recommended in patients with initially elevated CEA/CA19-9, owing to this therapy’s survival benefits compared to surgery first.” Please discuss this and other practical implications in greater detail.

Reply: Thanks very much for your excellent comments. The clinical implications of our findings indeed cover a broad topic, and it is important to have a thorough discussion on this matter. In the Discussion section, we have included additional relevant content addressing this issue as follows:

“Notably, elevated pre-NACT CEA/CA19-9 levels indicate a higher risk of death for GC patients, although this risk can be reduced by NACT. Normalization of post-NACT CEA/CA19-9 generates an effective therapeutic response, which can improve the OS of patients, while increased post-NACT CEA/CA19-9 value may predict an ineffective treatment. Then, the recommendation of NACT prior to surgery in patients with initially elevated CEA/CA19-9 levels is of significant importance due to the observed survival benefits compared to the strategy of surgery first. By administering chemotherapy before surgery, it is possible to target the tumor and potentially reduce its size, making it more amenable to surgical resection. This sequential treatment strategy has shown promise in improving patient outcomes. In addition to the specific recommendation for patients with elevated CEA/CA19-9 levels, there are other practical implications that should be considered. Firstly, the timing and duration of NACT should be carefully considered. It is important to strike a balance between achieving an adequate response to chemotherapy and minimizing the delay in surgical intervention. Another practical implication is the potential for individualized treatment approaches based on biomarker profiles. By analyzing a combination of biomarkers, including CEA and CA19-9, along with other relevant factors such as tumor stage and molecular characteristics, it may be possible to develop predictive models that can guide treatment decisions and improve patient outcomes.”

The section “2.7. Elevated CA19-9 predicted a higher probability of metastasis” is a separate study requiring proper description and discussion.

Reply: Thanks very much for your excellent comments. More details in section 2.7 and correlated discussion have been supplemented in the revised manuscript.

The Abstract does not properly reflect the main findings of the paper.

Reply: Thank you very much for your helpful suggestions. We have rewritten and modified the abstract to better reflect the main findings of our study.

The quality of English is absolutely sufficient for proper understanding of the paper, but some language editing is still required.

Reply: Thank you very much for your helpful suggestions. We apologize for any confusion caused by the language of our manuscript. In order to better describe the research findings and enhance the readability of the article, we have invited the assistance of a native English-speaking scholar to proofread and polish the language.

If you have any further questions, please feel free to contact us. We will do our best to improve this research.

Sincerely yours,

Xiaohuan Tang

Reviewer 2 Report

This study analyzed the prognostic value of changes in tumor markers CEA and CA19-9 following neoadjuvant chemotherapy (NACT) and D2 gastrectomy in advanced gastric cancer (AGC) patients. The authors found that AGC patients with elevated CEA and/or CA19-9 levels, both at diagnosis or after NACT, exhibited poor prognosis and shorter median overall survival (OS). It also presents that normalization of CEA and CA19-9 levels following NACT corresponded to better therapeutic outcomes and increased overall survival in AGC patients initially showing elevated CEA/CA19-9 levels. This study also found a link between elevated CEA/CA19-9 levels and the development of micro-metastases, indicated by RNA-seq data. Although it revealed potential importance from the cohort analysis, this study must include the novel finding in the manuscript.

First of all, the font size in the figures needs to be bigger to be easily legible. Unifying them to a size that ensures readability would be beneficial, especially in Fig 1, Fig 2, Fig 4, and Fig 5.

As described in the discussion, the authors recognized some limitations, such as its single-center design and imbalances in NACT regimens and cycles, which could affect the tumor markers' response. Therefore, further multi-center studies are suggested to validate these findings.

It did not explore the possible molecular mechanisms behind the changes in CEA and CA19-9 levels after NACT, which could limit the understanding of the observed correlations. As they speculate the micro-metastases from the RNA-seq data analysis, no data was shown in the present manuscript.

Minor point

In p2, lines 75-76, the numbers of patients at the clinic stage differ from that in Table 1. 

In p3, lines 85, the manuscript describes the average cycles for NACT, though Table 1 shows the median value for NACT.

Author Response

Reviewer 2

Comments and Suggestions for Authors

This study analyzed the prognostic value of changes in tumor markers CEA and CA19-9 following neoadjuvant chemotherapy (NACT) and D2 gastrectomy in advanced gastric cancer (AGC) patients. The authors found that AGC patients with elevated CEA and/or CA19-9 levels, both at diagnosis or after NACT, exhibited poor prognosis and shorter median overall survival (OS). It also presents that normalization of CEA and CA19-9 levels following NACT corresponded to better therapeutic outcomes and increased overall survival in AGC patients initially showing elevated CEA/CA19-9 levels. This study also found a link between elevated CEA/CA19-9 levels and the development of micro-metastases, indicated by RNA-seq data. Although it revealed potential importance from the cohort analysis, this study must include the novel finding in the manuscript.

Dear reviewer,

First, we would like to express our sincere thanks to you for the constructive and positive comments. We have read and addressed these helpful commons carefully and the details were listed point by point.

First of all, the font size in the figures needs to be bigger to be easily legible. Unifying them to a size that ensures readability would be beneficial, especially in Fig 1, Fig 2, Fig 4, and Fig 5.

Reply: We apologize for any confusion caused by the font size in the figures and thanks very much for your helpful suggestions. Then, we increased the font size of the figures to make them more readable. The modified figures have been added to the revised manuscript.

As described in the discussion, the authors recognized some limitations, such as its single-center design and imbalances in NACT regimens and cycles, which could affect the tumor markers' response. Therefore, further multi-center studies are suggested to validate these findings.

Reply: Thank you very much for your insightful comments. The validation of our study in multiple centers will indeed enhance its reproducibility. However, this study requires high data quality and completeness, and many centers did not have sufficient complete data, which is why data from other centers were not included. A prospective multicenter study is planned to address this issue.

It did not explore the possible molecular mechanisms behind the changes in CEA and CA19-9 levels after NACT, which could limit the understanding of the observed correlations. As they speculate the micro-metastases from the RNA-seq data analysis, no data was shown in the present manuscript.

Reply: Thank you very much for your excellent comments. CEA and CA19-9 are widely used traditional tumor markers in clinical practice. In the context of gastric cancer, CEA and CA19-9 are frequently used as prognostic and metastasis-related markers, although the precise molecular mechanisms underlying their associations are not fully understood. To better understand the potential mechanisms underlying the improved prognosis associated with the normalization of CEA/CA19-9, we characterized mRNA expression profiles of tumor tissues before NACT in 63 patients, including 50 and 13 patients with initial normal and elevated CEA/CA19-9, respectively. Functional enrichment analysis of DEGs identified by tissue RNA sequencing suggests that the survival benefits associated with the normalization of CEA/CA19-9 may be attributed to the clearance of potential micro-metastases by NACT. Detailed investigation of specific molecular pathways is crucial for further understanding the molecular mechanisms and the development of new therapeutic targets or enhancers. In future studies, we plan to delve deeper into these pathways to gain a better understanding and explore potential avenues for improved treatment strategies.

Thank you again for your helpful suggestions.

Minor point

In p2, lines 75-76, the numbers of patients at the clinic stage differ from that in Table 1.

Reply: Thanks for your excellent comments. We have corrected this error.

In p3, lines 85, the manuscript describes the average cycles for NACT, though Table 1 shows the median value for NACT.

Reply: Thanks for your excellent comments. We have modified the manuscript to use the median cycles of NACT instead of the average cycles to align with the description in Table 1.

The changes in tumor markers may be a common characteristic in many types of tumors, as the decrease in tumor marker levels after chemotherapy has also been reported in both lung cancer and pancreatic cancer.

Reply: Thanks very much for your excellent comments. We have reviewed serum marker studies in different tumor types and found a similar tendency in these markers, such as CEA, CA19-9, CA724 and CA-125, following chemotherapy. The details have been added in the introduction section as follows:

“Previous studies found that pretreatment CEA positive prior to NACT was an indicator for poor prognosis in LAGC patients and a significant decrease of tumor markers was observed after NACT. In addition, the changes in tumor markers may be a common characteristic in many types of tumors, as the decrease in tumor marker levels after chemotherapy has also been reported in pancreatic adenocarcinoma [18], advanced cholangiocarcinoma [19], locally advanced rectal cancer [20], ovarian cancer [21] and metastatic colorectal cancer [22, 23]. Although most studies suggest that a decrease in CA125 levels indicates a favorable response to chemotherapy in ovarian cancer [24], Coleman and colleagues discovered that serum CA125 concentration typically rises after the first and second cycles of chemotherapy and subsequently decreases [21]. In addition, Tsai et al. revealed that normalization of CA19-9 following NACT is a prognostic marker for survival in pancreatic adenocarcinoma [18]. However, the correlation between CEA/CA19-9 dynamic changes and the efficacy of NACT in LAGC has not been well analyzed.” In this study, our results revealed that normalization of CEA/CA19-9 is the strongest predictive marker for treatment response.

If you have any further questions, please feel free to contact us. We will do our best to improve this research.

Sincerely yours,

Xiaohuan Tang

Round 2

Reviewer 1 Report

-

Minor editing of English language required

Author Response

We greatly appreciate your recognition of the revised manuscript and our efforts. We have carefully reviewed the language issues and made necessary modifications to the manuscript based on the editor's suggestions.

Reviewer 2 Report

The authors very carefully made revisions in response to comments from both reviewers. 

The revised manuscript seems to have improved in quality.

The English problem in the manuscript has also been corrected, and I am happy to accept it.

Author Response

Dear reviewer,

We greatly appreciate your recognition of the revised manuscript and our efforts.  

Your valuable feedback has significantly helped us improve the content and quality of our research. 

Sincerely yours

Xiaohuan Tang